# Fast Estimation of Causal Interactions using Wold Processes

**Flavio Figueiredo**    **Guilherme Borges**    **Pedro O. S. Vaz de Melo**    **Renato Assunção**
Universidade Federal de Minas Gerais (UFMG)
{flaviovdf, guilherme.borges, olmo, assuncao}@dcc.ufmg.br
**Reproducibility:** http://github.com/flaviovdf/granger-busca

## Abstract

We here focus on the task of learning Granger causality matrices for multivariate point processes. In order to accomplish this task, our work is the first to explore the use of Wold processes. By doing so, we are able to develop asymptotically fast MCMC learning algorithms. With $N$ being the total number of events and $K$ the number of processes, our learning algorithm has a $O(N(\log(N) + \log(K)))$ cost per iteration. This is much faster than the $O(N^3 K^2)$ or $O(K^3)$ for the state of the art. Our approach, called GRANGER-BUSCA, is validated on nine datasets. This is an advance in relation to most prior efforts which focus mostly on subsets of the Memetracker data. Regarding accuracy, GRANGER-BUSCA is three times more accurate (in Precision@10) than the state of the art for the commonly explored subsets Memetracker. Due to GRANGER-BUSCA's much lower training complexity, our approach is the only one able to train models for larger, full, sets of data.

## 1   Introduction

In order to understand complex systems we need to know how their components (or entities) interact with each other. Networks (or graphs) offer a common language to model such systems, where their entities are represented by nodes and their interactions by edges [6]. The *networked point process* is a stochastic model for these systems, when data take the form of a time series of random events observed in each node. That is, in each node of a network we have a temporal point process, which is a random process whose realizations consist of the times $\mathcal{P} = \{t_j, \ j \in \mathbb{N}\}$ of isolated events. Networked point processes are probabilistic models designed to analyze the influence that events occurring in a node may have on the events occurring on other nodes of the network.

Recently, several fields used networked point processes to understand complex systems such as spiking biological neurons [36], social networks [8, 42] geo-sensor networks [22], financial agents in markets [37], television records [48] and patient visits [11]. One of the main objectives in these analyses is to uncover the causal relationships among the entities of the system, or the *interaction structure among the nodes*, which is also called the *latent network structure*. Typically, this is represented by a graph where edges connect nodes that affect each other and edge weights represent the intensity of this pairwise interaction. To the best of our knowledge, all methods that tackle this problem are based on Hawkes point process [25, 24] with a Granger causality framework [20] imposed to retrieve the causal graph from data [1, 48, 53, 35, 34]. A point process $\mathcal{P}_b$ is said to Granger cause another point process $\mathcal{P}_a$ when the past information of $\mathcal{P}_b$ can provide statistically significant information about the future occurrences of $\mathcal{P}_a$. We can thus encode causal relationships as a matrix [15, 16]. In a multivariate point process, this notion of causality can be reduced to measuring if the conditional intensity function of $\mathcal{P}_b$ is affected by the previous timestamps of $\mathcal{P}_a$ [48].

In contrast with the popular choice of using Hawkes process to model interacting processes, Wold processes [47] have been neglected as a possible model. Wold processes are a class of point processes

defined in terms of the joint distribution of inter-event times. Let $\delta_i = t_i - t_{i-1}$ be the waiting time for $i$-th event since the occurrence of the $(i-1)$-th event. The main characteristic of Wold processes is that the collection of inter-event times $\{\delta_i,\ i \in \mathbb{N}\}$ follows a Markov chain of finite memory. That is, different from Hawkes processes, whose intensity function depends on the whole history of previous events, the probability distribution of the $i$-th inter-event time $\delta_i$ depends only on the previous inter-event time $\delta_{i-1}$. When Wold processes are able to mimic the dynamics of complex systems [46, 45], this Markovian property can potentially boost the performance of learning algorithms as in our setting. Wold processes were proposed over sixty years ago, however, they remain scarcely explored in machine learning, which is unfortunate. As we will demonstrate in this paper, Wold processes can be both fast and accurate for some learning tasks.

We here present the first approach to tackle the discovery of latent network structures in point process data using Wold processes. Similar to recent efforts [1, 48], our goal in this work is to extract Granger causality [20] from multivariate point process data *only*. The main reason to consider the Wold process as an alternative to the Hawkes process is its Markovian structure. By adding Dirichlet priors over the mutual influences among the processes, we exploit the Markov property to develop learning algorithms that are asymptotically fast. We call our approach GRANGER-BUSCA. With $N$ being the number of observations in all processes and $K$ the number of processes, the state of the art approaches learn at a cost of $O(M\,N^3\,K^2)$ [48] ($M$ being defined by hyper-parameters) or $O(K^3)$ [1] per iteration. GRANGER-BUSCA, in contrast, learns at a cost of $O(N(\log(N) + \log(K)))$.

Equally important, our results show significant improvements over the state of the art methods. For instance, when we measure the Precision@10 score between our Granger causal estimates and the ground-truth number of mentions of the commonly explored Memetracker data [29], our results are at least three times more accurate than the most recent and most accurate method [1].

In summary, our main contributions are: (1) We present the first approach to extract Granger causality matrices via Wold processes; (2) We develop asymptotically fast algorithms to learn such matrices; (3) We show how GRANGER-BUSCA is much faster and more accurate than state of the art methods, opening up the potential of Wold processes for the machine learning community.

## 2    Background and Related Work

A temporal point process $\mathcal{P}$ is a probability model for a collection of times $\{0 \le t_0 < t_1 < t_2 < \ldots\}$ indexing the occurrences of random isolated events. Our context considers several point processes $\mathcal{P}_a, \mathcal{P}_b, \mathcal{P}_c, \ldots$ observed simultaneously, where each event is associated to a single point process: $\mathcal{P}_a = \{0 \le t_{a_0} < t_{a_1} < \ldots\}$. Let $\mathcal{P} = \bigcup_a \mathcal{P}_a$ be the union of all timestamps from all point processes with $N = |\mathcal{P}|$ being the total number of events. We denote by $N_a(t) = \sum_{i=1}^{|\mathcal{P}_a|} \mathbb{1}_{t_{a_i} \le t}$ the random cumulative count of the number of events up to (and including) time $t$ in process $\mathcal{P}_a$. The collection $\mathcal{N}_a = \{N_a(t) \mid t \in [0, T]\}$ is an equivalent representation of the point process $\mathcal{P}_a$.

The conditional intensity rate function is the fundamental tool for modeling and making inferences on point processes. Let $\mathcal{H}_a(t)$ be the random history of the process $\mathcal{P}_a$ up to, but not including, time $t$, called the *filtration*. Similarly, $\mathcal{H}(t)$ is defined as the filtration considering the collection $\mathcal{P}$ of all point processes. In the limit, as $dt \to 0$, the conditional intensity rate function is given by $\lambda_a(t|\mathcal{H}(t))dt = \Pr\left[N_a(t + dt) - N_a(t) > 0 \mid \mathcal{H}(t)\right] = \mathbb{E}\left(N_a(t + dt) - N_a(t) \mid \mathcal{H}(t)\right)$. The interpretation of this function is that, for a small time interval $dt$, the value of $\lambda_a(t|\mathcal{H}(t))\,dt$ is approximately equal to the expected number of events in $(t, t + dt]$. It can also be interpreted as the probability that process $\mathcal{P}_a$ has at least one event in the interval $(t, t + dt]$. The notation emphasizes that the conditional intensity at time $t$ depends on the random events that occurred previously.

The commonly explored Hawkes process $\mathcal{P}$ is defined by its set of conditional intensity functions:

$$\lambda_a(t|\mathcal{H}(t)) \quad = \mu_a + \sum_{b=0}^{K-1} \int_0^t \phi_{ba}(t - t')dN_b(t') = \mu_a + \sum_{b=0}^{K-1} \sum_{t_{b_i} < t} \phi_{ba}(t - t_{b_i}) \quad (1)$$

We can consider processes as being enumerated from $\{a, b, \cdots\} \in [0, K)$. $\mu_a$ captures the exogenous (Poissonian) background arrival rate of process $\mathcal{P}_a$. $\phi_{ba}$ captures the influence of the point process $\mathcal{P}_b$ on $\mathcal{P}_a$. $\phi_{ba}(t)$ has to be integrable, non-negative, and with $\phi_{ba}(t) = 0$ when $t < 0$. Usual forms of $\phi_{ba}(t)$ are the exponential and power-law functions [4]. With $\lambda_a(t)$ from Eq (1), there is evidence that $\mathcal{P}_b$ Granger causes $\mathcal{P}_a$ when there exists a time $t$ where $\phi_{ba}(t) \ne 0$ [48].

In contrast to the Hawkes process, a Wold process [14, 47] is defined in terms of a Markovian transition on the inter-event times (increments). Let $\mathcal{D}_a = \{\delta_{a_i} = t_{a_i} - t_{a_{i-1}}, \ldots\}$ be the stochastic process of time increments associated with point process $\mathcal{P}_a$. The Markovian transition between increments is established by the transition probability density $\Pr[x = \delta_{a_{i-1}} \mid y = \delta_{a_i}]$ which measures the likelihood of $\delta_{a_i}$ given the value of the previous increment $\delta_{a_{i-1}}$ [47, 14].

It is important to point out that, for most forms of $\Pr[x \mid y]$, the model is analytically intractable [21]. However, when $\Pr[x \mid y]$ has an exponential form, such as $\Pr[x \mid y] = f(x)e^{-f(x)y}$, the model has several interesting properties [12, 13]. In this particular form, the next increment is exponentially distributed with rate $\lambda = f(x)$ where $x$ is the previous increment, i.e.: $\delta_{i+1} \sim Exponential(\lambda = f(\delta_i))$. For the particular case of $f(x) = \beta + \alpha x$, the work of Cox [12] and Daley [13, 14] derive the stationary distribution of increments showing that it is of the form: $\Pr[x] \propto (\beta + \alpha x)^{-1} e^{-\alpha x}$.

GRANGER-BUSCA is motivated by recent efforts [2, 45, 46] that employ variations of the exponential Wold process (defined above). Instead of defining the Wold process in terms of its interval exponential rate, such efforts defined the process in terms of the conditional mean $\mu(x) = \mathbb{E}_a(\delta_{a_i} | \delta_{a_{i-1}} = x) = \beta + \alpha x$ of an exponentially distributed random variable. This class of point processes is called *self-feeding processes* (SFP). For the particular case of $\alpha = 1$ and $\beta = \text{median}(\delta_{a_i})/e$, with $e \approx 2.718$ being the Euler constant, [45, 46] showed that the stationary behavior can be very well approximated by the more tractable Log-Logistic distribution. This new form of specifying a Wold process is interesting because it is able to capture both exponential and power-law behavior, disparate behaviors simultaneously observed in real data [2, 45, 46]. Realizations of this process tend to generate bursts of intense activity followed by long periods of silence.

*Busca* [2] is another point process model based on Wold processes and it is GRANGER-BUSCA's starting point. Starting from a SFP model with $\alpha = 1$, the authors accommodate the less frequent long periods of waiting times observed in some real datasets, the process adds a background Poissonian rate ($\mu_a$). The conditional intensity can thus be derived given by:

$$\lambda_a(t \mid \mathcal{H}_a(t)) = \mu_a + \frac{1}{\beta + \delta_{a_p}}, \tag{2}$$

where $\delta_{a_p} = t_{a_p} - t_{a_{p-1}}$ with $p$ being defined by $\arg\max\{p : t_{a_i} < t\}$. That is, $p$ the index associated to the closest previous event to $t$. $\delta_{a_p}$ is thus the previously observed increment.

Over the years, several Hawkes methods have been created for different applications with varying asymptotic costs [5, 8, 9, 10, 33, 41]. We now discuss the methods most related to GRANGER-BUSCA. Achab et at. [1] presents a $O(K^3)$ approach. Xu et al. [48] learns kernels at cost of $O(M\,K^2\,N^3)$. Here, $M$ represents the number of basis functions used to approximate the kernel no-parametrically. Similarly Yang et al. [49] discusses a non-parametric Hawkes that does not explore the infinite memory. The authors show that after $W$ events, the kernel is adequately estimated. However, the proposed method still scales in the order of $O(M\,K^3\,W)^1$, $M$ represents again a pre-defined number of functions used to approximate the kernel, $W$ is the maximum number of previous events to be considered. A similar proposal is presented by Etesami et al. [18]. In contrast, GRANGER-BUSCA has a computation complexity of $O(N(\log(N) + \log(K)))$ per iteration. We achieve this low cost by employing a Bayesian inference where at each step the algorithm learns, for each event in $\mathcal{P}$, which other process, if any, caused that event. Past efforts have employed similar sampling approaches on Hawkes based models [17, 34, 35] which again do not scale.

It is important to point out that Isham [27] was one of the first **and few** to discuss multivariate Wold processes. However, the author was mostly focused on analytical properties of the Multivariate Wold Process on a very specific setting (an infinite process defined on the unit circle).

## 3  Model

We define GRANGER-BUSCA using Figure 1, which exemplifies the behavior of the model with two processes. In (a), we show the events of each process as a horizontal line of symbols, triangles in

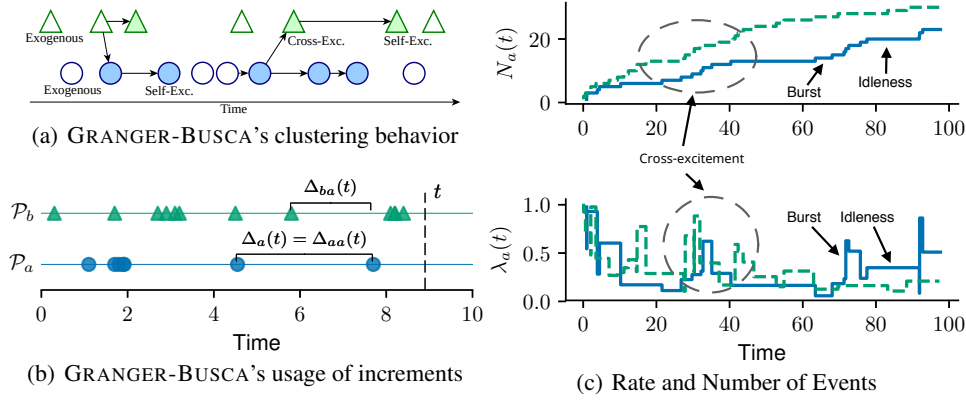

(a) GRANGER-BUSCA's clustering behavior

(b) GRANGER-BUSCA's usage of increments

(c) Rate and Number of Events

Figure 1: GRANGER-BUSCA at work. Plot (a) shows the events of process $\mathcal{P}_a$ (circles) and process $\mathcal{P}_b$ (triangles). The arrows show the excitement component of the model. Plot (b) illustrates how $\Delta_{aa}(t)$ and $\Delta_{ba}(t)$ are calculated. Plot (c) shows the cumulative random processes $N_a(t)$ and $N_b(t)$ in the top, while the bottom plot shows the random conditional intensity functions $\lambda_a(t)$ and $\lambda_b(t)$.

the upper row for process $\mathcal{P}_b$, and circles in the bottom row for process $\mathcal{P}_a$. The unfilled symbols represent events that are caused in an exogenous way, not triggered by any other event. We shall detail how to label events in Section 4. The filled symbols are events that appear as a causal influence from some other previous event. We say that these events have been *excited* by the previous occurrence of other events. The directed edge connects the origin event to the resulting event. Edges crossing the two parallel lines of events represent the *cross-excitement* component. In this case, one event in a given process stimulates the occurrence of events in another process. Another situation is due to *self-excitement*, when events in a given process stimulate further events in the same process. These are represented by the horizontal arrows in Figure 1(a). Figure 1(c) shows the behavior of the random $N_a(t)$ and the random conditional intensity function $\lambda_a(t)$. Notice that $\lambda_a(t)$ behaves like a step function. That is, the rate of arrivals is fixed until the next arrival. The figure also shows the burst-idleness cycle observed with GRANGER-BUSCA, as well as the cross-excitation.

To formalize GRANGER-BUSCA, let us first redefine $\delta_{a_p}$ as a function $\Delta_a(t)$. Recall that $p$ is the index of the previously observed event in $\mathcal{P}_a$ that is closest to $t$. Also recall that $\delta_{a_p}$ is the last observed increment. The function notation will simplify our extension to a multivariate Wold process. See Figure 1(b) for an illustration. Let us now define define $q$ as the event in $\mathcal{P}_b$ that is closest to $t_{a_p}$, that is $\arg\max\{q : t_{b_q} < t_{a_p} < t\}$. With such an index, we can denote by $\Delta_{ba}(t)$ the difference between $t_{a_p}$ and $t_{b_q}$, i.e.: $\Delta_{ba}(t) = t_{a_p} - t_{b_q}$. When the expected values of $\Delta_{ba}(t)$ are small, events in $\mathcal{P}_b$ usually precede $\mathcal{P}_a$. In this sense, one intuition on how GRANGER-BUSCA works is that larger observed values of $\Delta_{ba}(t)$ lead to weaker evidence for the influence of process $\mathcal{P}_b$ on process $\mathcal{P}_a$.

The above behavior motivates GRANGER-BUSCA's multivariate conditional intensity function:

$$\lambda_a(t) = \underbrace{\mu_a}_{\text{Exogenous Poisson Rate}} + \underbrace{\sum_{b=0}^{K-1} \frac{\alpha_{ba}}{\beta_b + \Delta_{ba}(t)}}_{\text{Endogenous Wold Rate}} \quad (3)$$

The random intensity function $\lambda_a(t)$ is the sum of two components. The first one is $\mu_a$ and it represents the exogenous events in process $\mathcal{P}_a$ that are generated at a Poissonian constant rate $\mu_a$ by unit of time. The other component is a sum over all processes, including the same $a$ process. The terms in this sum represent the increment on the baseline rate $\mu_a$ contributed by other previous events from $\mathcal{P}_a$ itself (self-excitement) or from other processes (cross-excitement). Based on a very sparse representation, the entire history is concentrated only on the most recent time gaps between events. Hence, the process $\mathcal{P}_b$ influence on process $\mathcal{P}_a$ is represented by the ratio $\alpha_{ba}/(\beta_b + \Delta_{ba}(t))$. The numerator is a scale factor measuring the amount of cross-excitement: when is equal to zero, there is no influence from $\mathcal{P}_b$ on $\mathcal{P}_a$. The denominator models how this cross-excitement takes place. At time $t$, we add the time gap $\Delta_{ba}(t)$ to the flat value $\beta_b$. A large gap $\Delta_{ba}(t)$ makes the contribution of

the ratio $\alpha_{ba}/(\beta_b + \Delta_{ba}(t))$ small relative to the baseline rate $\mu_a$. Otherwise, a small gap raises this contribution up to its maximum possible contribution rate of $\alpha_{ba}/\beta_b$.

**Model parameters and definitions:** $\Theta = \{G, \beta, \mu\}$ contains the full set of model parameters. $G = [\alpha_{ba}]$ is the Granger matrix of the model. We require that $\alpha_{ba} \geq 0$ and that $\sum_{a=0}^{K-1} \alpha_{ba} = 1$. Hence, the value $\alpha_{ba}$ in the $G[b, a]$ is proportional to the fraction of events from process $\mathcal{P}_b$ that triggered events on $\mathcal{P}_a$. By definition, $G$ is row stochastic [26]. This property leads to several interesting characteristics of the model, that we further develop throughout the rest of this section. The vector $\beta = [\beta_b]$ captures the the overall influence strength for each process $\mathcal{P}_b$. That is, when a process influences another, it does so by exponentially distributed inter-event times with a mean of $\beta_b + \Delta_{ba}(t)$. As we now show, to guarantee stationary conditions, it is necessary that $1 \leq \beta_b < \infty$. Finally, we consider that each event $t_{a_i}$ has a *latent* label indicating either that it is exogenous or which process caused it (edges in Figure 1(a)). Estimation of $\Theta$ would be trivial if these labels were known. Thus, our learning algorithm (see Section 4) focuses on estimating on such labels from data.

As pointed out, GRANGER-BUSCA's stationarity (some authors also call this property stability [14, 34]) depends on $1 \leq \beta_b < \infty$. Let $B(t)$ be the diagonal matrix where each diagonal cell is equal to $1/(\beta_b + \Delta_{ba}(t))$. Moreover, let $\Phi(t) = B(t)G = [\frac{\alpha_{ba}}{\beta_b + \Delta_{ba}(t)} = \lambda_{ba}(t)]$. Each value in $\Phi(t)$ is the cross-intensity for a pair of processes. If $\mathcal{P}_b$ are represented in the rows and $\mathcal{P}_a$ in the columns, the expected number of events at an infinitesimal region for $\mathcal{P}_a$ is equal to the row sum of this matrix. This value is simply GRANGER-BUSCA's cross-feeding intensity without the exogenous factor. Now, let $||X||$ be the induced $l_\infty$-norm (maximum row sum).

**Definition 1 (Stationarity):** Notice that, $||\Phi(t)|| < 1$ [26]. The proof for this property is straightforward and has interesting implications. As $G$ is row-stochastic, we have $||G|| = 1$. Also, $\beta_b + \Delta_{ba}(t) > 1$ since $\beta_b \geq 1$ and $\Delta_{ba}(t) > 0$. The matrix $B(t)$ will either scale down this norm or keep it unchanged. The spectral radius is $\rho(\Phi(t)) < 1$. Also, $\lim_{k \to \infty} \Phi(t)^k = 0$.

Due to the above definitions, the model is stationary/stable as it will never generate infinite offspring. In fact, the total number of offspring at any given time is determined by the sum $\sum_{k=0}^{\infty} \Phi(t)^k = (\mathbb{I} - \Phi(t))^{-1}$. Next, we explore the definition of Granger causality for point processes [16, 48].

**Definition 2 (Granger Causality):** A process $\mathcal{P}_a$ is said to be independent of any other process $\mathcal{P}_b$ when: $\lambda_a(t|\mathcal{H}_a(t)) = \lambda_a(t|\mathcal{H}_\mathcal{P}(t))$ for any $t \in [0, T]$. In contrast, $\mathcal{P}_b$ is defined to Granger cause $\mathcal{P}_a$ when: $\lambda_a(t|\mathcal{H}_a(t)) \neq \lambda_a(t|\mathcal{H}_\mathcal{Q}(t))$, where $\mathcal{Q} = \mathcal{P}_a \cup \mathcal{P}_b$. As a consequence, given two processes $\mathcal{P}_a$ and $\mathcal{P}_b$ whose intensity functions follow Eq (3), Granger causality arises when $\alpha_{ba} \neq 0$ [2]

## 4 Learning GRANGER-BUSCA

Recall from Figure 1 that GRANGER-BUSCA exhibits a cluster like behavior. That is, exogenous observations arrive at a fixed rate, with each observation being able to trigger new observations leading to a burst/idleness cycle. Based on this remark, we developed our Markov Chain Monte Carlo (MCMC) sampling algorithm to learn GRANGER-BUSCA from data. Our algorithm will work by sampling, for each observation $t_{a_i}$, the hidden process (or label), that likely caused this observation. In other words, we sample a latent variable, $z_{a_i}$, which takes a value of $b \in [0, K-1]$ when process $\mathcal{P}_b$ influences $t_{a_i}$. When the stamp is exogenous, we set this value to a constant $K$. Such a number merely represents a label (exogenous) and does not affect our sampling.

To simplify our learning strategy, we decided to fix $\beta = 1$. We notice that such a value shown to be sufficient for GRANGER-BUSCA to outperform state of the art baselines (see Section 5). Later in this section, we shall discuss that our learning algorithm is easily adaptable for general forms of the intensity function. $G$ is estimated based on the number of events that $\mathcal{P}_b$ caused on $\mathcal{P}_a$, whereas $\mu$ is estimated as the maximum likelihood rate for a Poisson process based on the exogenous events for $\mathcal{P}_a$. We can thus learn GRANGER-BUSCA with an Expectation Maximization approach. Hidden labels are estimated in the Expectation step. The matrix $G$ can also be readily updated in such a step. With the labels, $\mu$ estimated in the maximization step. Next, we discuss our fitting strategy.

Initially, we explored the Markovian nature of the process to evaluate $\Delta_{ba}(t)$ at a $O(\log(N))$ cost. Given some labels' assignments for the events, we obtain $\Delta_{ba}(t)$ with a binary search over $\mathcal{P}_a$ and $\mathcal{P}_b$.

We explain now how we update the $z_{a_i}$ labels. Given any event at $t_{a_i}$ in process $\mathcal{P}_a$, the event either exogenous or induced by some other previous event on $\mathcal{P}_a$ or from some other process $\mathcal{P}_b$. By the superposition theorem [14, 34, 35, 17], we obtain the conditional probability that an individual event at $t_{a_i}$ is exogenous or was caused by process $\mathcal{P}_b$ (where $b$ can be equal to $a$ for SFP like behavior) as:

$$\Pr[t_{a_i} \in \text{Exog.}] = \frac{\mu_a}{\mu_a + \sum_{b'=0}^{K-1} \lambda_{b'a}(t_{a_i})}, \qquad \Pr[t_{a_i} \leftarrow \mathcal{P}_b] = \frac{\lambda_{ba}(t_{a_i})}{\mu_a + \sum_{b'=0}^{K-1} \lambda_{b'a}(t_{a_i})}, \quad (4)$$

where the $\leftarrow$ operator indicates causality. Notice that, $z_{a_i} = b$ is equivalent to $t_{a_i} \leftarrow \mathcal{P}_b$. Eq (4) is carried out conditionally on the history $\mathcal{H}_a(t_{a_i})$.

We can accelerate substantially our evaluations by using a binary modified Fenwick Tree [19] (the F+Tree [51]) data structure if we break the $z_{a_i}$ label assignment into two steps. First, we decide if it is exogenous. Given it is not, we select the inducing process based on the conditional probability:

$$\Pr[t_{a_i} \leftarrow \mathcal{P}_b \mid t_{a_i} \notin \text{Exog.}] = \frac{\lambda_{ba}(t_{a_i})}{\sum_{b'=0}^{K-1} \lambda_{b'a}(t_{a_i})}. \tag{5}$$

The evaluation of the probability that an event is not exogenous has an $O(1)$ cost because we estimate $\Pr[t_{a_i} \notin \text{Exog.}] \triangleq 1 - e^{-\mu_a(t_{a_i} - t_{\mu_a})}$ where $t_{\mu_a}$ is the last event before $t_{a_i}$ that arrived from an exogenous factor[3]. This probability is the complement of the probability that zero Poissonian events happened between $t_{\mu_a}$ and $t_{a_i}$. As $G$ is row stochastic, we first add a Dirichlet prior over each row of this matrix ($\alpha_p$). We finally sample the hidden labels $z_{a_i}$ as follows:

1. For each process $\mathcal{P}_a$
    (a) Sample row $a$ from $G$ as $\sim Dirichlet(\alpha_p)$
2. For each process $\mathcal{P}_a$
    (a) For each observation $t_{a_i} \in \mathcal{P}_a$
        i. Sample $p \sim Uniform(0, 1)$
            A. When $p < e^{-\mu_a(t_{a_i} - t_{\mu_a})}$
                $z_{a_i} \leftarrow$ exogeneous
            B. Otherwise
                Sample $z_{a_i} \sim Multinomial(Eq\ 5)$

Model parameters are estimated through a MCMC sampler. Starting at an arbitrary random state (i.e., labels' assignment), let $n_{ba}$ be the number of times $\mathcal{P}_b$ influenced $\mathcal{P}_a$. Similarly, $n_b$ captures the number of times $\mathcal{P}_b$ influenced any process, including itself. The conditional probability of hidden labels for every observation, $\mathbf{z}$, is given by: $\Pr[\mathbf{z} \mid \Theta] = \prod_{a=0}^{K-1} \prod_{i=0}^{|\mathcal{P}_a|-1} \Pr[t_{a_i} \notin \text{Exog.}] \times \Pr[t_{a_i} \leftarrow \mathcal{P}_b \mid t_{a_i} \notin \text{Exog.}]$. By factoring $G$, labels are sampled based on the collapsed Gibbs sampler [43]. Thus, we re-write Eq (5) below. In this next equation, we point out the Proposal and Target Distributions used on the Metropolis Based Sampler (discussed next).

$$\Pr[t_{a_i} \leftarrow \mathcal{P}_b \mid t_{a_i} \notin \text{Exog.}] \propto \overbrace{\underbrace{\frac{n_{ba}^{-t_{a_i}} + \alpha_p}{n_b^{-t_{a_i}} + \alpha_p K}}_{\text{Proposal Distribution}} \frac{1}{\beta_b + \Delta_{ba}(t)}}^{\text{Target Distribution}}, \tag{6}$$

where $\alpha_{ba}^{-t_{a_i}} = (n_{ba}^{-t_{a_i}} + \alpha_p)/(n_b^{-t_{a_i}} + \alpha_p K)$ being the current estimate of $\alpha_{ba}$, with $n_{ba}^{-t_{a_i}}$ being the count for the pair $n_{ba}$ excluding the current assignment for $t_{a_i}$ and $n_b^{-t_{a_i}}$ being similarly defined. Thus, the full algorithm follows an EM approach. After labels are assigned in the Expectation step, we can compute $\mu_a$ for every process by simply estimating the maximum likelihood Poissonian rate. Given that it takes $O(\log(N))$ time to compute $\Delta_{ba}(t)$ and $O(K)$ time to compute Eq 5, the total sampling complexity per learning iteration for the Gibbs sampler will be of $O(N \log(N) K)$.

**Speeding Up with a Metropolis Based Sampler:** The $K$ factor in the traditional Gibbs sampler may be reduced by exploiting specific data structures, such as the AliasTable [52, 32] or the F+Tree [51]. In order to speed-up GRANGER-BUSCA, we shall employ the latter (F+Tree). The trade-offs between

the two are discussed in [51]. Our choice is motivated by the fact that the F+Tree does not make use of stale samples. Thus, we can perform multinomial sampling with a $O(\log(K))$ cost. Given that the AliasTable cannot be updated quickly, it is usually only suitable at later iterations [52, 32].

We exploit the F+Tree by changing our sampler to a Metropolis Hasting (MH) approach. Using the common notation for an MH, let $Q(b)$ be the proposal probability density function as detailed in Eq 6. Here, $b$ is a candidate process which may replace the current assignment $c = z_{a_i}$. When the target distribution function is simply Eq 6, i.e., $P(c) = Eq\ 6$, we can sample the assignment $z_{a_i}$ using the acceptance probability $\min\{1, (P(c)Q(b))/(P(b)Q(c))\}$. In other words, at each iteration we either keep the previous $z_{a_i}$ or replace with $b$ according to the acceptance probability. As discussed, with the F+Tree, we can sample from $Q(b)$ in $O(\log(K))$ time. We can also update the tree with the new probabilities after each step with the same cost. Given that F+Tree has a $O(K \log(K))$ cost to build, we build the tree once per process. Finally, as required for the Metropolis Hasting algorithm, it is trivial to see that the proposal distribution is proportional to the target distribution [23].

**Parallel Sampler:** With the F+Tree, candidates are sampled at a $O(\log(K))$ cost per event. Moreover, finding previous stamp costs $O(\log(N))$. By adding these two costs, the algorithmic complexity of GRANGER-BUSCA per iteration is $O(N(\log(N) + \log(K)))$. Finally, notice that the sampler is easy to parallelize. That is, by iterating over events on a per-process basis, we parallelize the algorithm by scheduling different processes to different CPU cores. Overall, only vector of variables is shared across processes ($n_b$ in Eq (6)). In our implementation, each core will read $n_b$ for each process before an iteration. After the iteration, the value is thus updated globally. Our sampler falls in the case of being Asynchronous with Shared Memory as discussed in [44].

**Learning different Kernels** Consider the equivalent rewrite of $\lambda_a(t) = \mu_a + \sum_{b=0}^{K-1} \alpha_{ba}\omega_{ba}(t)$, where, $\omega_{ba}(t) = 1/(\beta_b + \Delta_{ba}(t))$ for GRANGER-BUSCA in particular. With this new form, the model acts as a mixture of intensities ($\omega_{ba}(t)$). Each pairwise intensity is weighted by the causal parameters $\alpha_{ba}$. Now, notice that using this form our EM algorithm is easily extensible for different functions $\omega_{ba}(t)$. The E-step is able to estimate the causal graph (considering that Eq (6) $\hat{=} \alpha_{ba}^{-t_{a_i}}\omega_{ba}(t)$). The M-Step provides maximum likelihood estimates for the specific parameters appearing in $\omega_{ba}(t)$. In fact, even unsupervised forms may be learned. As we discuss in the next section, we keep the aforementioned intensity given that it is simpler and outperforms baselines in our datasets.

## 5   Results and Experiments

We now present our main results. GRANGER-BUSCA is compared with three recently proposed baselines methods: ADM4 [53], Hawkes-Granger [48] and Hawkes-Cumulants [1]. The code for each method is publicly available via the library `tick`[4]. Experiments were performed on a dedicated Azure Standard DS15 v2 instance with 20 Intel Xeon CPU E5-2673 v3 cores and 140GB of memory. We point out that we perform comparisons are performed with Hawkes methods given that it is the most prominent framework. There is no Wold method for our task. We compare with approaches that are: parametric [53] and non-parametric [48, 1], and explore finite [1] and infinite [53, 48] memory.

**Hyper Parameters:** ADM4 enforces an exponential kernel and with a fixed rate. We employ a Tree-structured Parzen Estimator to learn such a rate [7], optimizing for the best model in terms of log-likelihood. For Hawkes-Granger, we fit the model with $M = 10$ basis functions as suggested in [1]. Finally, Hawkes-Cumulants [1] also has a single hyper parameter called the *half-width*, which was also estimated using [7]. Training is performed until convergence or until 300 iterations is reached. Our MCMC sampler executes for 300 iterations with $\alpha_p = \frac{1}{K}$ and $\boldsymbol{\beta} = \mathbf{1}$.

**Datasets:** We evaluate GRANGER-BUSCA and the aforementioned three baselines on 9 different datasets, all of which were gathered from the Snap Network Repository[5]. Out of the nine datasets, we pay particular attention to the Memetracker data, which is the only one used to evaluate all past efforts. The Memetracker dataset consists of web-domains linking to one another. We pre-process the Memetracker dataset using the code made available by [1]. The other eight datasets consist of source nodes (e.g., students or blogs) sending events to destination nodes (e.g., messages or citations). Each datasets can be represented as triples $\mathcal{D} = \{(source, destination, timestamp)\}$. The ground truth network is defined as the graph $\mathcal{G}_t = \{\mathcal{V}_t, \mathcal{E}_t, \mathcal{W}_t\}$, where the vertices $\mathcal{V}_t$ are both the sources

Table 1: Datasets used for Experiments and Precision Scores for Full Datasets. Due to their sizes, only GRANGER-BUSCA is able to execute in all datasets. To allow comparisons, we execute baselines methods with only the Top-100 destination nodes. Other results are presented in Table 2 and Figure 2.

| | # Proc (K) | # Obs. (N) | N (Top-100) | Span | %NZ | P@5 | P@10 | P@20 | TT(s) |
|---|---|---|---|---|---|---|---|---|---|
| bitcoinalpha [28] | 3,257 | 23,399 | 2,279 | 5Y | 0.2% | 0.26 | 0.14 | 0.07 | 3 |
| bitcoinotc [28] | 4,791 | 33,766 | 2,328 | 5Y | 0.1% | 0.25 | 0.14 | 0.07 | 7 |
| college-msg [39] | 1,313 | 58,486 | 10,869 | 193D | 1.1% | 0.36 | 0.30 | 0.19 | 1 |
| email [31, 50] | 803 | 327,677 | 92,924 | 803D | 3.74% | 0.23 | 0.28 | 0.32 | 4 |
| sx-askubuntu [40] | 88,549 | 879,121 | 58,142 | 7Y | 0.006% | 0.25 | 0.13 | 0.06 | 2774 |
| sx-mathoverflow [40] | 16,936 | 488,984 | 59,602 | 7Y | 0.07% | 0.28 | 0.16 | 0.09 | 98 |
| sx-superuser [40] | 114,623 | 1,360,974 | 64,866 | 7Y | 0.006% | 0.26 | 0.14 | 0.07 | 4614 |
| wikitalk [30, 40] | 251,154 | 7,833,140 | 211,344 | 6Y | 0.003% | 0.25 | 0.14 | 0.07 | 27540 |
| memetracker-100 [29] | 100 | 24,665,418 | - | 9M | 9.85% | 0.30 | 0.29 | 0.22 | 114 |
| memetracker-500 [29] | 500 | 39,318,989 | - | 9M | 4.44% | 0.30 | 0.30 | 0.23 | 274 |

Table 2: Comparing GRANGER-BUSCA (GB) with Hawkes-Cumulants (HC) Memetracker.

| | Precision@5 | | Precision@10 | | Precision@20 | | Kendall | | Rel. Error | | TT(s) | |
|---|---|---|---|---|---|---|---|---|---|---|---|---|
| | HC | GB | HC | GB | HC | GB | HC | GB | HC | GB | HC | GB |
| top-100 | 0.06 | **0.30** | 0.09 | **0.29** | 0.01 | **0.22** | 0.05 | **0.26** | 1.0 | **0.44** | **87** | 114 |
| top-500 | 0.01 | **0.30** | 0.01 | **0.30** | 0.02 | **0.23** | 0.08 | **0.20** | 1.8 | **0.06** | 715 | **274** |

and the destinations. Edges, $e = (b, a) \in \mathcal{E}_t$ and $\{b, a\} \subseteq \mathcal{V}_t$, represent the relationship between two entities. The weighted adjacency matrix of this graph, $\mathbf{G}_t$, is our ground-truth. It is defined as: $\mathbf{G}_t[b, a] = (\# (b, a) \in \mathcal{D})/(\# b \text{ is a source})$. To compose each process from these datasets, each *destination* node represents a process. In compliance to our notation, we call such processes $\mathcal{P}_a$. Notice that we do not consider *source* nodes, $\mathcal{P}_b$, as processes. Thus, the models will aim to extract causal graphs based on the likelihood that reception of messages at a $\mathcal{P}_a$ destination will trigger incoming messages. If this is the case, we have evidence that $\mathcal{P}_a$ is *also* be a source node. Finally, we pre-process the data to only consider destinations that have also sent messages.

**Metrics:** We evaluate GRANGER-BUSCA and its competitors using the Precision@n, Kendall Correlation and Relative Error Scores. Each score is measured per node (or row of $\mathbf{G}$), and is summarized for the entire dataset as the average over every node. Both the Kendall Correlation, as well as the Relative Errors, were proposed as evaluation metrics for networked point processes in [1]. Precision@n captures the average fraction of edges in $\mathbf{G}$ out of the top $n$ edges ordered by weight which are also present in $\mathbf{G}_t$. As we shall discuss, there are several limitations with the Kendall and Relative Error scores due to graph sparsity. We argue that Precision@n measured at different cut-offs ($n$) is a more reliable evaluation metric for large and sparse graphs, as the ones we explore here.

**Results:** Table 1 describes the datasets used in this work presenting their number of nodes and of observations. Most baselines do not execute on large datasets in under 24h of training time (TT), in contrast with GRANGER-BUSCA. Given the asymptotic complexity, we estimate that some models may take months to execute. Hence, to allow comparisons with GRANGER-BUSCA, we considered subsamples containing only the events involving the Top-100 destinations. We pay particular attention to Top-100 and 500 nodes for Memetracker, given that it was explored in prior efforts [53, 48, 1].

Table 1 also presents the Precision@n scores for the GRANGER-BUSCA. Recall that ours is the **only** approach able to execute on the full sets of data. Firstly, notice that the Kendall and Relative Error scores are absent from Table 1. Given that datasets are sparse – as shown by the fraction of non-zeros in the ground truth, or %NZ, in Table 1 – the Kendall Correlations and Relative Errors are unreliable metrics for large networks. It is well known that complex networks as ours have long-tailed distributions for the edge-weights [6], leading to the high sparsity levels (%NZ) seen in Table 1. With most cells being zeros, Kendall Scores also tend to zero as most sources connect to few destinations. Similarly, the relative errors will likely increase. In order to avoid divisions by zero, previous efforts impose a constant penalization, of one, when zero edges exist between two nodes. Similar to the Kendall Correlation, this penalization will also dominate the score due to the sparsity.

Because of the above limitations of prior efforts and metrics, we are unable to present a fair comparison with competitors on the full datasets. To achieve this goal, in Table 2, we present the overall scores for GRANGER-BUSCA and the Hawkes-Cumulants (HC) [1] approach, focusing only on the Memetracker data. In this setting, Hawkes-Cumulants has already been shown to be more accurate and faster than

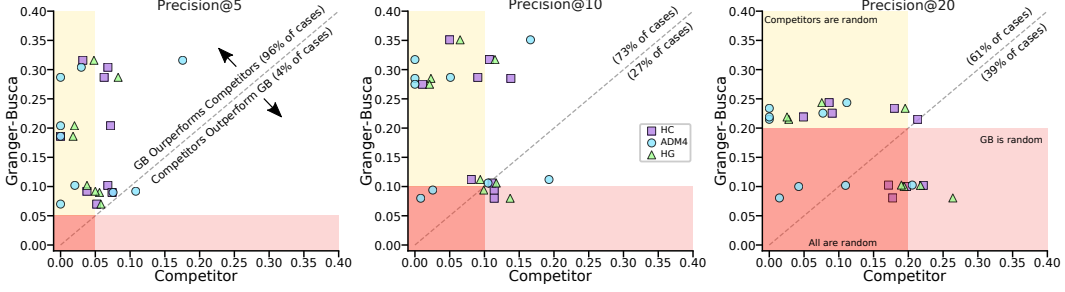

Figure 2: Precision Scores for the Top-100 datasets.

ADM4 [53] and Granger-Hawkes [48] (GH). Observe that GRANGER-BUSCA is more accurate than the competing method in every score. Indeed, Precision@n scores are at least three times greater depending on the cut-off (Precision@5, 10 and 20). Kendall Scores also show substantial gain (250%), with GRANGER-BUSCA achieving 0.20 and HC achieving 0.08 correlations on average. Relative errors for GRANGER-BUSCA are also 56% lower than HC (1.0 versus 0.44). Finally, notice how GRANGER-BUSCA is slightly slower than HC when fewer nodes are present (100), but significantly surpasses HC in speed as $K$ increases. This comes from the $O(K^3)$ cubic cost incurred by HC.

To present an overall view of GRANGER-BUSCA against all three competing methods (ADM4, HC and HG), in Figure 2 we show Precision@5, 10 and 20 scores for each approach on every Top-100 dataset. A total of 26 (out of 27 possible) points are plotted in the figure. One single setting, HC with Memetracker, is absent since the model was unable o train sufficient time. The $x$-axis of the figure presents the Precision@n score for the baseline. Similarly, the $y$-axis shows the Precision@n score for GRANGER-BUSCA. We also show three regions where either GRANGER-BUSCA or competitors perform worse than a Null model. This model was created by performing uniformly random rankings. Notice that for Precision@5 and Precision@10, GRANGER-BUSCA outperforms most baselines on a large fraction of the datasets. In fact, for Precision@5, there is only one setting where ADM4 outperforms GRANGER-BUSCA. As the precision cut-off increases, so does the efficacy of the Null model (i.e., it easier for a random ranking to recover top edges). For Precision@20, there does exists some cases where GRANGER-BUSCA is outperformed by baseline methods. However, the majority of these cases exist in the region where both models are below the efficacy of a Null model.

**Why does the model work?** Recall that a Wold process is an adequate model when there is a strong dependency between consecutive inter-event times $\delta_t$ and $\delta_{t+1}$. To explain our results, we measured the correlation between $\delta_t$ and $\delta_{t+1}$ for pairs of interacting processes from the ground-truth data. Out of nine datasets, the worst case had the median Pearson correlation per pair equal to 0.3, a moderate value. However, in the remaining eight datasets this median was above 0.70, indicating the adequacy of a Wold model. The high linear dependency implies that $\delta_{t+1} \approx \alpha\delta_t + \beta \rightarrow \mathbb{E}[\delta_{t+1}] \approx \alpha\,\mathbb{E}[\delta_t] + \beta$. Thus, $\mathbb{E}[\delta_{t+1}]$ is a linear function, $f$, of the previous inter-event, justifying the intensity: $\delta_{t+1} \sim Exponential(\mu = f(\delta_t))$ (see Section 4 for a discussion on how to learn other forms).

It is also important to understand the limitations of non-parametric methods such as HC. Recall that HC relies on the statistical moments (e.g., mean/variance) of the inter-event times [1]. Given that web datasets exhibit long tails (with theoretical distributions exhibiting high, or even infinite, variance), such moments will not accurately capture the underlying behavior of the dataset (see Section 2).

# 6  Conclusions and Future Work

In this work, we present the first method to extract Granger causality matrices via Wold Processes. Though it was proposed over sixty years ago, this framework of point processes remain largely unexplored by the machine learning community. Our approach, called GRANGER-BUSCA, outperforms recent baseline methods [1, 48, 53] both in training time and in overall accuracy. GRANGER-BUSCA works particularly well when extracting the top connections of a node (Precision@5, 10, 20).

Given the efficacy of GRANGER-BUSCA, our hope is that current results may open up a new class of models, Wold processes, to be explored by the machine learning community. Finally, GRANGER-BUSCA can be used to explore real world behavior in complex large scale datasets.

## Acknowledgements

We thank Fabricio Murai and the anonymous reviewers for providing comments. We also thank Gabriel Coutinho for discussions on the mathematical properties of GRANGER-BUSCA, as well as Alexandre Souza for providing pointers to prior studies. This work has been partially supported by the project ATMOSPHERE (atmosphere-eubrazil.eu), funded by the Brazilian Ministry of Science, Technology and Innovation (Project 51119 - MCTI/RNP 4th Coordinated Call) and by the European Commission under the Cooperation Programme, Horizon 2020 grant agreement no 777154. Funding was also provided by the authors' individual grants from CNPq, CAPES and Fapemig. Computational resources were provided by the Microsoft Azure for Data Science Research Award (CRM:0740801).

## Footnotes

[1]The authors discuss a $O(K^2)$ cost for a **parallel** algorithm with hyper-parameters being $O(1)$. While this is correct asymptotically, we compare, for all methods, the non-parallel cost with hyper-parameters. In practice, hyper-parameters have a multiplicative effect on learning time. Moreover, for every parallel method (GRANGER-BUSCA included), the reduction factor of $K$ ($K^2$ from $K^3$) is only possible via unbounded parallelization (one process per CPU), being unfeasible for large data: $K >> \#CPUs$.

[2]When interpreting results from GRANGER-BUSCA, we relax the above condition of strict independence to $\alpha_{ba} \approx 0$. In such cases, we can state that we have no statistical evidence for Granger causality.

[3]The $\triangleq$ operator means *equality by definition*.

[4]`https://github.com/X-DataInitiative/tick`. Results produced with version 0.4.0.0.

[5]`https://snap.stanford.edu/data/`

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

## A  Simulating GRANGER-BUSCA

In Algorithm 1 we show how Ogata's Modified Thinning algorithm [38] is adapted for GRANGER-BUSCA. We initially point out that some care has to be taken for the initial simulated timestamps. Given that $t_{a_i}$ (the previous observation) does not exist, the algorithm will need to either start with a synthetic initial increment of fall back to the Poisson rate. In the algorithm, the rate of each individual process is computed. Then, a new observation is generated based on the sum of such rates. Given that each process will behave like a Poisson process while a new event does not surface (Figure 1), the sum of these processes is also a Poisson process. Lastly, we employ a multinomial sampling to determine the process from which the observation belongs to.

## B  Log Likelihood

We now derive the log likelihood of GRANGER-BUSCA for parameters $\mathbf{\Theta} = \{\boldsymbol{G}, \boldsymbol{\beta}, \boldsymbol{\mu}\}$. For a point process with intensity $\lambda(t)$, the likelihood can be computed as [14]:

$$L(\mathbf{\Theta}) = \prod_{i=i}^{N} \lambda(t_i)\, exp(-\int_0^t \lambda(t)dt). \tag{7}$$

Considering the intensity of each process separately, we can write the log likelihood as:

$$LL_a(\mathbf{\Theta}) = \sum_{t_{a_i} \in \mathcal{P}_a} \left( log\big(\lambda_a(t_{a_i})\big) \right) - \int_0^t \lambda_a(t)dt \tag{8}$$

$$= \sum_{t_{a_i} \in \mathcal{P}_a} \left( log\big(\mu_a + \sum_{b=0}^{K-1} \frac{\alpha_{ba}}{\beta_b + \Delta_{ba}(t_{a_i})}\big) \right) - T_a\mu_a - \sum_{t_{a_i} \in \mathcal{P}} \sum_{b=0}^{K-1} \frac{\alpha_{ba}(t_{a_i} - t_{a_{i-1}})}{\beta_b + \Delta_{ba}(t_{a_{i-1}})}$$

Here, $T_a$ is the last event from $\mathcal{P}_a$. The expansion of the integral $\int_0^{T_a} \lambda_a(t)dt$ comes from the stepwise nature of $\lambda_a(t)$ (see Figure 1). For simplicity, let us initially assume that there is only one process. As discussed in the paper, computing $\Delta_{ba}(t)$ has a $log(N)$ cost. Due to summations of the form, $\sum_{t_i \in \mathcal{P}} \sum_{b=0}^{K-1}$, the cost to evaluate $LL_a(\mathbf{\Theta})$ is $O(K\,N\,log(N))$. $N \log(N)$ is the cost to evaluate $\Delta_{ba}(t)$ for every observation.

