[Supplementary Material]


[44] A. Terenin and E. P. Xing. Techniques for proving asynchronous convergence results for markov chain monte carlo methods. In *NIPS*, 2017.

[45] P. O. S. Vaz de Melo, C. Faloutsos, R. Assunção, R. Alves, and A. A. Loureiro. Universal and distinct properties of communication dynamics: how to generate realistic inter-event times. *ACM Transactions on Knowledge Discovery from Data (TKDD)*, 9(3), 2015.

[46] P. O. S. Vaz de Melo, C. Faloutsos, R. Assunção, and A. Loureiro. The self-feeding process: A unifying model for communication dynamics in the web. In *WWW*, 2013.

[47] H. Wold. On stationary point processes and markov chains. *Scandinavian Actuarial Journal*, 1948(1-2), 1948.

[48] H. Xu, M. Farajtabar, and H. Zha. Learning granger causality for hawkes processes. In *ICML*, 2016.

[49] Y. Yang, J. Etesami, N. He, and N. Kiyavash. Online learning for multivariate hawkes processes. In *NIPS*, 2017.

[50] H. Yin, A. R. Benson, J. Leskovec, and D. F. Gleich. Local higher-order graph clustering. In *KDD*, 2017.

[51] H.-F. Yu, C.-J. Hsieh, H. Yun, S. Vishwanathan, and I. S. Dhillon. A scalable asynchronous distributed algorithm for topic modeling. In *WWW*, 2015.

[52] J. Yuan, F. Gao, Q. Ho, W. Dai, J. Wei, X. Zheng, E. P. Xing, T.-Y. Liu, and W.-Y. Ma. Lightlda: Big topic models on modest computer clusters. In *WWW*, 2015.

[53] K. Zhou, H. Zha, and L. Song. Learning social infectivity in sparse low-rank networks using multi-dimensional hawkes processes. In *AISTATS*, 2013.

## A   Simulating GRANGER-BUSCA

In Algorithm 1 we show how Ogata's Modified Thinning algorithm [38] is adapted for GRANGER-BUSCA. We initially point out that some care has to be taken for the initial simulated timestamps. Given that $t_{a_i}$ (the previous observation) does not exist, the algorithm will need to either start with a synthetic initial increment of fall back to the Poisson rate. In the algorithm, the rate of each individual process is computed. Then, a new observation is generated based on the sum of such rates. Given that each process will behave like a Poisson process while a new event does not surface (Figure 1), the sum of these processes is also a Poisson process. Lastly, we employ a multinomial sampling to determine the process from which the observation belongs to.

## B   Log Likelihood

We now derive the log likelihood of GRANGER-BUSCA for parameters $\Theta = \{G, \beta, \mu\}$. For a point process with intensity $\lambda(t)$, the likelihood can be computed as [14]:

$$L(\Theta) = \prod_{i=i}^{N} \lambda(t_i) \, exp(-\int_0^t \lambda(t)dt). \tag{7}$$

Considering the intensity of each process separately, we can write the log likelihood as:

$$LL_a(\Theta) = \sum_{t_{a_i} \in \mathcal{P}_a} \left( log\big(\lambda_a(t_{a_i})\big) \right) - \int_0^t \lambda_a(t)dt \tag{8}$$

$$= \sum_{t_{a_i} \in \mathcal{P}_a} \left( log\big(\mu_a + \sum_{b=0}^{K-1} \frac{\alpha_{ba}}{\beta_b + \Delta_{ba}(t_{a_i})}\big)\right) - T_a\mu_a - \sum_{t_{a_i} \in \mathcal{P}} \sum_{b=0}^{K-1} \frac{\alpha_{ba}(t_{a_i} - t_{a_{i-1}})}{\beta_b + \Delta_{ba}(t_{a_{i-1}})}$$

Here, $T_a$ is the last event from $\mathcal{P}_a$. The expansion of the integral $\int_0^{T_a} \lambda_a(t)dt$ comes from the stepwise nature of $\lambda_a(t)$ (see Figure 1). For simplicity, let us initially assume that there is only one process. As discussed in the paper, computing $\Delta_{ba}(t)$ has a $log(N)$ cost. Due to summations of the form, $\sum_{t_i \in \mathcal{P}} \sum_{b=0}^{K-1}$, the cost to evaluate $LL_a(\Theta)$ is $O(K\,N\,log(N))$. $N \log(N)$ is the cost to evaluate $\Delta_{ba}(t)$ for every observation.

**Algorithm 1** Ogata's Thinning Algorithm Adapted for GRANGER-BUSCA

> **Input:** max time $T$, num proc $K$, $\boldsymbol{G}$, $\boldsymbol{\beta}$, $\boldsymbol{\mu}$
> **Output:** observations $\mathcal{P}$
> $t \leftarrow 0$
> $\mathcal{P} = \{\}$
> $\boldsymbol{\lambda} = zeros(K)$
> $\boldsymbol{n} = zeros(K)$
> **while** $t < T$ **do**
>     {Compute the rate for each process using $\boldsymbol{G}$, $\boldsymbol{\beta}$, $\boldsymbol{\mu}$. The rate $\lambda_a(t)$ depends of $t_{a_p}$ and $t_{b_q}$ to compute $\Delta_{ba}(t)$ (see Eq (3)). If such timestamps do not exist, fall back to a Poisson process, that is: $\lambda_a(t) = \mu_a$}
>     **for** $a \leftarrow 0$ **to** $K - 1$ **do**
>         $\boldsymbol{\lambda}[a] \leftarrow \lambda_a(t)$
>     **end for**
>     {Move forward in time. That is, sample a new observation with rate $sum(\boldsymbol{\lambda}))$}
>     Sample $dt \sim Exponential(\lambda = sum(\boldsymbol{\lambda}))$
>     $t \leftarrow t + dt$
>     {Sample a process $a$ to such that $t_{a_i} = t$}
>     Sample $u \sim Uniform(0, sum(\boldsymbol{\lambda}))$
>     $a \leftarrow 0$
>     $c \leftarrow 0$
>     **while** $a < K - 1$ **do**
>         $c \leftarrow c + \boldsymbol{\lambda}[i]$
>         **if** $c \geq \boldsymbol{\lambda}[i]$ **then**
>             **break**
>         **end if**
>         $a \leftarrow a + 1$
>     **end while**
>     $i \leftarrow \boldsymbol{n}[a]$
>     $t_{a_i} \leftarrow t$
>     $\mathcal{P} \leftarrow \mathcal{P} \cup \{t_{a_i}\}$
>     $\boldsymbol{n}[a] \leftarrow \boldsymbol{n}[a] + 1$
> **end while**

Now, let us return to the case of multiple processes. Let $N_a$ be the number of events for process $a$. Next, $M$ is the number of events in the processes with the most of such a number. That is, $M = max(N_a \mid \forall a)$. The cost of $LL(\boldsymbol{\Theta})$, naively, will be of $O(K\,N\,log(M))$. This comes from the summation: $\sum_{a=0}^{K-1} LL_a(\boldsymbol{\Theta}) = K \log(M) \sum_{a=0}^{K-1} N_a$. To simplify the comparison with past methods, in our manuscript we did not detail our runtime cost in terms of $M$. Strictly speaking, our fitting algorithm with the MCMC sampler performs at a cost of: $O(N\,(\log(M) + \log(K)))$.

## C  Fitting Algorithm

The algorithm is shown in Algorithm 2, with the E-step being detailed in Algorithm 3. The maximization step, for GRANGER-BUSCA in particular, is a MLE estimation for a Poisson process. The pseudo-code shown here is not parallel and builds the F+Tree naively. By updating $n_b$ using a sloppy counter (see Chapter 11 of [3]) across processing cores, one only needs to iterate over $\mathcal{P}_a$ to compute $n_{ba}$. The counter consists of a local count of $n_b$ for each processor. After a certain number of steps, say at every $x$-iterations, $n_b$ is synced with a master parameter server.

The runtime of the algorithm may be optimized by either pre-computing or caching $\Delta_{ba}(t_{a_i})$ for every observation from every process. Nevertheless, this pre-computation comes at a memory cost of $O(N\,K)$ being likely is prohibitive for larger datasets. We can however cache a small subset of such values to amortize the $O(\log(N))$ cost down to $O(1)$ for cache hits. Secondly, the $O(\log(K))$ cost can also be amortized with an AliasTable. With these two optimizations, it is possible to implement optimized versions of the sampling algorithm that execute at a $O(N)$ amortized cost per iteration.

**Algorithm 2** Sampling GRANGER-BUSCA

---

**Input:** all observations $\mathcal{P}$, prior $\alpha_p$, num. iter $I$
**Output:** $\boldsymbol{G}$, $\boldsymbol{\mu}$

$K \leftarrow |\mathcal{P}|$
$\mathcal{Z} \leftarrow \{\}$
$\mu \leftarrow Zeros(K)$

{Sample initial state from a random uniform $\in [0, K]$. The value $K$ is reserved to indicate exogeneous events. $IsPoisson(z_{a_i})$ **returns** $z_{a_i} = K$.}
**for** $a \leftarrow 0$ **to** $K - 1$ **do**
    $\mathcal{Z}_a \leftarrow \{\}$
    **for** $i \leftarrow 0$ **to** $|\mathcal{P}_a| - 1$ **do**
        $z_{a_i} \leftarrow UniformInt(0, K + 1)$      {$z_{a_i} \in [0, K]$}
    **end for**
    $\mathcal{Z}_a \leftarrow \mathcal{Z}_a \cup \{z_{a_i}\}$
**end for**

{Sample hidden labels}
**for** $iter \leftarrow 0$ **to** $I - 1$ **do**
    **for** $a \leftarrow 0$ **to** $K - 1$ **do**
        $EStep(\mathcal{P}_a, \mathcal{Z}, \alpha_p, \boldsymbol{\mu}[a], K)$
        $\boldsymbol{\mu}[a] \leftarrow MStep(\mathcal{P}_a, \mathcal{Z}_a)$
    **end for**
**end for**

$\boldsymbol{G} \leftarrow Zeros(K, K)$
{Compute Output. $\boldsymbol{G}[b, a] = \frac{n_{ba} + \alpha_p}{n_b + \alpha_p K}$}
**return** $\boldsymbol{G}$, $\boldsymbol{\mu}$

---

**Algorithm 3** Expectation Step ($EStep$)

---

**Input:** observations $\mathcal{P}_a$, current state $\mathcal{Z}$, prior $\alpha_p$, num proc. $K$, exogeneous rate $\mu_a$
{The tree is populated with the probability that each process $\mathcal{P}_b$ can cause $t_{a_i}$. i.e., $\boldsymbol{t}[b] = \frac{n_{ba} + \alpha_p}{n_b + \alpha_p K}$}
$\boldsymbol{t} \leftarrow FPTreeBuild(\mathcal{Z})$

**for** $i \leftarrow 0$ **to** $|\mathcal{P}_a| - 1$ **do**
    **if not** $IsPoisson(z_{a_i})$ **then**
        $b \leftarrow z_{a_i}$
        $\boldsymbol{t}[b] \leftarrow \frac{n_{ba} + \alpha_p - 1}{n_b + \alpha_p K - 1}$
    **end if**

    **if not** $Uniform(0, 1) < e^{-\mu_a(t_{a_i} - t_{\mu_a})}$ **then**
        $z_{a_i} \leftarrow K$
    **else**
        $c \leftarrow z_{a_i}$
        $b \leftarrow FPTreeSample(\boldsymbol{t})$
        {**See Eq** (6) **for the proposal** $Q$ **and target** $P$}
        **if** $Uniform(0, 1) < \min\{1, (P(c)Q(b))/(P(b)(c))\}$ **then**
            $z_{a_i} \leftarrow b$
        **end if**
        $\boldsymbol{t}[b] \leftarrow \frac{n_{ba} + \alpha_p + 1}{n_b + \alpha_p K + 1}$
    **end if**
**end for**

---