[Reviews · NeurIPS 2018]

Reviewer 1



This paper focuses on learning Granger causality in Wold processes. Accordingly, a fast estimation approach based on MCMC is proposed. Quality: This paper extends learning Granger causality in Hawkes point process to Wold processes, for the reason that Wold process has the Markovian structure. The simulation results are not complete. The authors claim that they compare with other three methods, but in Table 1 only the results from the proposed method are showed. Table 2 only shows the results compared with Hawkes-Cumulants, but not with the other two. Clarity: Overall, the presentation is ok. Originality: The novelty of this paper is limited. It is an application of Granger causality to Wold process.

Reviewer 2



Summary: This work studies the Granger causality in a special type of point processes called Wold processes. This type of processes is defined in terms of a Markovian transition on the inter-event times. The Markovian transition between increments is established by the transition density functions. The intensity of the point processes considered in this work consists of two components. The first one is constant value that represents the exogenous events in the process and the other one is a summation of rational terms that capture the influences of the other processes including self-excitements. Learning the Granger causality in such processes boils down to learning the parameters of the second component. This work proposes an MCMC-based Algorithm for learning these parameters called Granger-Busca. The complexity of the proposed algorithm is O(N(log N + log K)), where N being the total number of events and K the number of processes. They also validate their algorithm’s performance via several experiments. Pros: Studying the Granger causality in Wold point processes is novel and can be interesting to several disciplines. The proposed algorithm is faster that the state of the art (for Multivariate Hawkes processes). Cons: The proposed method is designed for a special sub-class of Wold processes with particular form of the intensity function, Equation (3). Unlike the proposed methods for Hawkes processes, the proposed method in this work cannot be extended to processes more general forms of the transition functions. The comparison between the complexity of the proposed method and the previous algorithms for Hawkes is not a fair comparison, since there are dealing with different classes of point processes with different complexity. For instance, as also mentioned in the paper, Hawkes processes do not have finite memory. The MemeTracker dataset was previously modeled via Hawkes processes and in this work using Busca processes. What is the justification for this modeling? What is their model validation criterion? It seems the authors merely refer the work of Bacry and Song groups. Some very relevant references are missing. For instance: Learning network of multivariate hawkes processes: A time series approach Etesami et al UAI 2016. Online learning for multivariate Hawkes processes Yang et al NIPS 2017. Hawkes Process Modeling of Adverse Drug Reactions with Longitudinal Observational Data Bao et al. Machine Learning and Healthcare 2017. The multivariate hawkes process in high dimensions: Beyond mutual excitation Chen et al. ArXiv Overall, this interesting work can be strengthened by generalizing the proposed algorithm to a broader class of transition functions. I rate this work as “marginally above acceptance” because their algorithm reduces the complexity extensively compared to the state of the art. However, as I mentioned the state of the art algorithms are considering Hawkes processes that have infinite memory.

Reviewer 3



Summary: The paper introduces a new method, called Granger-Busca, to learn (Granger) causal structure from spiking time-series data based on Wold processes instead of the more commonly used Hawkes point processes. A key difference is that the latter requires the whole history of previous events to model the distribution of the next event per point, whereas the Wold process only needs to keep track of the time since the last event per point. If this can provide a suitable approximation to the actual generating process, the inference procedure becomes much more efficient, scaling up much better than standard state-of-the-art approaches. The parameters of the model can be learned using a straightforward MCMC sampling scheme to infer which point processes are likely to influence/trigger which others. Further speedups can be achieved by changing to a MH approach and/or employ parallel sampling of multiple processes. The resulting method is evaluated on a number of different (de facto) benchmark data sets, and shown to significantly outperform current state-of -the-art in terms of both speed and accuracy. Although I am no expert on learning causal interactions between point processes, the approach presented in the paper seems both novel and very promising. As far as I could tell the Granger-Busca model and associated techniques to learn/infer the model parameters are sound, and should be applicable to a wide range of real-world problems. The paper itself is well written, and does a reasonable attempt to guide readers less familiar with the subject through the various steps and principles involved. The evaluation shows some impressive performances, however, it also raises a few question marks. For example, I can understand why the method is faster than certain Hawkes based versions, however, it is less clear why it should also be much more accurate. Given the extremely low precision rates achieved by Hawkes-Cumulants on the Memetracker data in Tab/Fig2 it suggests that this particular application is in fact not suitable for HC, which also makes the comparison a bit of a straw man, in turn removing some of the gloss of the results. Still, the overall impression remains favourable. As a result, I would expect that the ideas in this paper are of potential interest to many other researchers in the field, and may inspire further developments and improvements in this challenging application domain, and hence recommend accept.

Reviewer 4



Summary: This work studies the Granger causality in a special type of point processes called Wold processes. This type of processes is defined in terms of a Markovian transition on the inter-event times. The Markovian transition between increments is established by the transition density functions. The intensity of the point processes considered in this work consists of two components. The first one is constant value that represents the exogenous events in the process and the other one is a summation of rational terms that capture the influences of the other processes including self-excitements. Learning the Granger causality in such processes boils down to learning the parameters of the second component. This work proposes an MCMC-based Algorithm for learning these parameters called Granger-Busca. The complexity of the proposed algorithm is O(N(log N + log K)), where N being the total number of events and K the number of processes. They also validate their algorithm’s performance via several experiments. Pros: Studying the Granger causality in Wold point processes is novel and can be interesting to several disciplines. The proposed algorithm is faster that the state of the art (for Multivariate Hawkes processes). Cons: The proposed method is designed for a special sub-class of Wold processes with particular form of the intensity function, Equation (3). Unlike the proposed methods for Hawkes processes, the proposed method in this work cannot be extended to processes with more general form of the transition functions. The comparison between the complexity of the proposed method and the previous algorithms for Hawkes is not a fair comparison, since there are dealing with different classes of point processes with different complexity. For instance, as also mentioned in the paper, Hawkes processes do not have finite memory. The MemeTracker dataset was previously modeled via Hawkes processes and in this work using Busca processes. How do the authors justify such modeling? What is their model validation criterion? Overall, this interesting work can be strengthen by generalizing the proposed algorithm for broader class of transition functions. I rate this work as “marginally above acceptance” because their algorithm reduces the complexity extensively compared to the state of the art. However, as I mentioned the state of the art algorithms are considering Hawkes processes that have infinite memory.